# When, How, and to What Extent Are Individuals with Unresponsive Wakefulness Syndrome Able to Progress? Neurobehavioral Progress

**DOI:** 10.3390/brainsci11010126

**Published:** 2021-01-19

**Authors:** Enrique Noé, Joan Ferri, José Olaya, María Dolores Navarro, Myrtha O’Valle, Carolina Colomer, Belén Moliner, Camilla Ippoliti, Anny Maza, Roberto Llorens

**Affiliations:** 1NEURORHB, Servicio de Neurorrehabilitación de Hospitales Vithas, Fundación Vithas, Callosa d’En Sarrià 12, 46007 València, Spain; joan@neurorhb.com (J.F.); olaya@neurorhb.com (J.O.); loles@neurorhb.com (M.D.N.); myrtha@neurorhb.com (M.O.); carol@neurorhb.com (C.C.); belen@neurorhb.com (B.M.); camila@neurorhb.com (C.I.); rllorens@i3b.upv.es (R.L.); 2Neurorehabilitation and Brain Research Group, Instituto de Investigación e Innovación en Bioingeniería, Universitat Politècnica de València, Camino de Vera s/n, 46011 València, Spain; amaza@i3b.upv.es

**Keywords:** unresponsive wakefulness syndrome, vegetative state, minimally conscious state, disorders of consciousness, brain damage, predictors, recovery, mortality

## Abstract

Accurate estimation of the neurobehavioral progress of patients with unresponsive wakefulness syndrome (UWS) is essential to anticipate their most likely clinical course and guide clinical decision making. Although different studies have described this progress and possible predictors of neurobehavioral improvement in these patients, they have methodological limitations that could restrict the validity and generalization of the results. This study investigates the neurobehavioral progress of 100 patients with UWS consecutively admitted to a neurorehabilitation center using systematic weekly assessments based on standardized measures, and the prognostic factors of changes in their neurobehavioral condition. Our results showed that, during the analyzed period, 34% of the patients were able to progress from UWS to minimally conscious state (MCS), 12% of the total sample (near one third from those who progressed to MCS) were able to emerge from MCS, and 10% of the patients died. Transition to MCS was mostly denoted by visual signs, which appeared either alone or in combination with motor signs, and was predicted by etiology and the score on the Coma Recovery Scale-Revised at admission with an accuracy of 75%. Emergence from MCS was denoted in the same proportion by functional communication and object use. Predictive models of emergence from MCS and mortality were not valid and the identified predictors could not be accounted for.

## 1. Introduction

Severe brain injuries can lead to prolonged periods of impaired consciousness, which are clinically referred to as disorders of consciousness (DOC). Currently, the term DOC encompasses the unresponsive wakefulness syndrome (UWS), formerly known as vegetative state (VS), which is characterized by the presence of wakefulness in absence of awareness [1], and the minimally conscious state (MCS), which is characterized by the presence of minimal or inconsistent awareness [2,3]. However, the clinical view of DOC has evolved significantly since the description of the VS by Jennet and Plum back in 1972 [4]. The theories of functional segregation that were believed to explain the nature of cerebral functionality until well into this century supported the prevailing dichotomist concept of consciousness. According to it, patients could be either conscious or unconscious/vegetative, based on the presence or absence of observable clinical signs of interaction with the environment, respectively. Consequently, and promoted by the mediatisation of some cases [5], recovery of consciousness was interpreted as a magical or miraculous event. The inconceivable chances for recovery of consciousness by that time could have turned diagnosis into a self-fulfilling prophecy. According to it, those patients with the worst clinical condition could have received less therapeutic attention, which entails serious clinical and ethical issues [6]. The consensus reached by the Aspen Neurobehavioral Conference Workgroup on describing the MCS as a distinguishable clinical condition from the VS at the end of the 1990s [7,8] and the definition of behavioral diagnostic criteria in 2002 [9] meant a revolution in the clinical interpretation of DOC. The delineation of the MCS began to shape the current conception of recovery of consciousness as a gradual process, which fitted better with the emerging theories of functional integration that were supported by the latest neuroimaging findings on brain connectivity by that time [10]. A few years later, the European Task Force on Disorders of Consciousness coined the term UWS to refer the VS avoiding the negative connotations of the term [11].

As our understanding of the neural basis of consciousness and the clinical management of the medical complications associated with patients with DOC improve, survival rates have more than doubled [12,13], and an increasing number of studies has documented recovery of consciousness even years after the onset [14,15,16,17,18,19,20,21,22]. Although the neurobehavioral progress of each patient with a DOC is highly distinctive [23], knowing the general prognosis of patients with a comparable neurologic condition is essential to anticipate the most likely clinical course in each case. Prognostic information is of major importance to guide clinical decision making [24], which can be particularly delicate in DOC as, in the most vulnerable cases, it can lead to end-of-life decisions [25]. Establishing accurate prognosis is, therefore, especially relevant for patients with UWS because they have shown to have worse outcome in comparison to patients in MCS [8,26,27,28,29]. 

A large part of the studies that described the clinical evolution of patients with UWS prior to the description of the MCS were included in the report that the Multi Society Task Force group published in 1994, which served as a prognostic reference for many years [1]. According to this report, the likelihood of neurobehavioral improvement in the first year after the injury was 52% for traumatic cases and 15% for non-traumatic cases. However, the reliability of these figures has been questioned because they include retrospective old series of patients that could not represent the current state of the clinical practice [13]. Since then, a large number of studies have described the neurobehavioral progress of patients with UWS, often showing large discrepancies between studies, which can be partially explained by differences in the methodologies and included samples [23,26,27,29,30,31,32,33,34,35,36,37,38,39,40,41,42,43,44,45,46,47,48,49,50,51]. Some studies have also investigated demographic, clinical and neurophysiological predictors of the neurobehavioral progress in patients with DOC, with a particular emphasis on patients with UWS. Factors such as age, etiology, injury location and the baseline time since injury, neurobehavioral condition, and level of disability have been associated with neurobehavioral outcomes of patients with DOC [29,33,34,35,36,37,38,41,46,47,49]. Neurophysiological metrics as well as responses to neuroimaging active and passive paradigms, have been also shown to be associated with patients’ prognosis [23,34,39,40,49,50].

Although all the existing studies provide valuable information about the neurobehavioral progress of patients with UWS, they incorporate small samples [26,27,30,31,32,35,37,47,49,50], include infrequent assessments [23,29,33,34,37,38,40,41,42,50], do not use standardized neurobehavioral assessment measures [26,27,43] (contrary to updated recommendations [3,44]), focus on patients with specific etiologies [31,36,39,50], combine data with those from patients with MCS [23,33,34,35,36,37,38,39,41,46,47,49], and, except on rare occasions [37,42,45], provide information about behavioral signs. In addition, with regards to the studies on predictors, most of the existing studies described to date do not provide the parameters of the models [23,29,33,36,37,38,39,41,49], and more importantly, have an uncertain predictive value, probably optimistic, as performance is determined with the same data used to develop the predictive model [23,29,33,35,37,38,39,46,47,50]. Instead, a prognostic model should be validated with data not used in the development process and, preferably, extracted from different clinical settings [52]. While the medical records of patients with UWS are limited and clinical procedures differ across clinical facilities, models should be, at least, internally validated with resampling methods, such as cross validation or bootstrapping. A recent international multicenter collaboration by the International Brain Injury Association’s DOC Special Interest Group is noteworthy for overcoming this recurring methodological limitation [34]. However, the study include data from patients either with UWS or in MCS who may or may not be undergoing rehabilitation and are followed for less than six months. 

These limitations might hinder accurate estimation and prognosis of the neurobehavioral progress of patients with UWS at admission of rehabilitation facilities, which might be essential to adjust expectations and improve decision-making. The objective of this study was, therefore, to investigate the neurobehavioral progress of a representative cohort of patients with UWS who were provided multidisciplinary rehabilitation through systematic weekly assessments based on standardized measures, and the prognostic factors of neurobehavioral improvement and mortality. The functional independence of this cohort of patients is described in an accompanying paper [53].

## 2. Materials and Methods

### 2.1. Participants

The demographic and clinical data of all the patients admitted from January 2004 to January 2020 to the inpatient neurorehabilitation program of a network of four hospitals were retrospectively analyzed. Patients were included if they had had a brain injury of any etiology between one and 12 months prior to admission, if they were diagnosed as with UWS according to the Coma Recovery Scale-Revised (CRS-R) at the moment of admission, and if their neurobehavioral condition was clinically monitored for a minimum of 12 months since the injury, or until emergence from MCS or decease. Patients were excluded if their neurobehavioral condition was not assessed in two consecutive weeks.

The study was approved by Comité Ético de Investigación Clínica del Hospital Clínic Universitari de València (2019002). Written informed consent to participate in the study was obtained from the legal representative of all patients.

### 2.2. Procedure

The admitting diagnosis of all the patients was made considering the score on the Spanish version CRS-R [47,54] in a minimum of five assessments during the first week after admission and the cut-off scores proposed for this scale [47]. 

Patients were administered daily sessions of physical therapy and multimodal stimulation customized to their particular needs. The medical monitoring of the patients focused on avoiding and treating possible medical complications that could be caused by their physical condition and prolonged immobility (i.e., passive range-of-motion exercises, postural care, daily sitting, etc.), reducing agitation by using beta blockers (i.e., propanolol, atenolol, etc.) or neuroleptics (i.e., quetiapine, olanzapine, etc), relieving pain by using analgesics (i.e., acetaminophen, nonsteroidal anti-inflammatory drugs, etc.), as well as promoting the recovery of consciousness by using drugs (i.e., amantadine, zolpidem, etc.), multisensory stimulation (i.e., interactive projections, aroma therapy, musical selections, tactile stimulation, etc.), and non-invasive brain stimulation (i.e., transcranial direct current stimulation or transcutaneous auricular vagus nerve stimulation). 

The neurobehavioral condition of all the patients was assessed weekly with the CRS-R until they were discharged, died, or emerged from MCS. Assessments were conducted during the morning, from 10 a.m. to 12 a.m., by a trained neuropsychologist. In case of decease, the causes of the death were collected. 

### 2.3. Data Analysis

Comparisons of continuous demographic and clinical variables were performed using Student t-tests. Chi-square were used to compare categorical variables.

Multivariable binary logistic regressions were performed to examine possible predictors of neurobehavioral progress, evidenced as either a transition from UWS to MCS or an emergence from MCS, and death. A stratified k-fold cross-validation methodology was used to identify potential predictors of the final models and determine their performance. This methodology is especially interesting in cases of limited sample, as it enables testing and validating the models using all the available data by means of repeated resampling, which maximizes the total number of cases used for testing and, potentially, helps to protect against overfitting. In summary, cross-validation randomly regroups the available dataset in “k” subsets or folds. All the subsets but one are used to build or train a model and the remaining subset is used to validate the trained model. This way it is possible to estimate how well the model will generalize to new data, which have not been used to train the model. The training and validation subsets change “k” times iteratively so, eventually, all subsets are used for training and validation. However, the important thing is that for each iteration, the validation subset is never used to train the model and, therefore, represents new data for the trained model. 

For each of the three predictive models of interest (transition from UWS to MCS, emergence from MCS and mortality) the procedure was as follows. First, the available data (100 patients) were divided into 5 subsets (20 patients each) with equal distribution of the dependent variable. In each of five iterations, 4 subsets (80 patients) were used to train a model and the remaining subset (20 patients) were used to validate it. Univariate logistic regressions were conducted for each demographic (age, sex, education) and clinical variables (etiology, time since injury, and total score on the CRS-R) at admission to detect potential predictors. All the variables that showed a *p* < 0.1 in the univariate analyses were considered potential candidates and were introduced as independent variables in stepwise, backward selection, multivariable logistic regressions. It should be noted that these regressions did not represent the final model but were performed to estimate its performance. The multivariable regressions were performed with a threshold of *p* = 0.1 for removal and with a maximum iterations set at 20. The multivariable regression models were evaluated using both the training and validation subsets. The classifier cutoff were selected using the optimal threshold of the Receiver Operating Characteristic (ROC) curves. These thresholds were defined as the first interaction between the ROC curve and a straight line whose slope was the cost of misclassifying a positive class as a negative one, and vice versa. The same classifier cutoff was used for the validation of the model. Accuracy, precision, sensitivity (or recall), specificity and the area under the ROC curve (AUC) were used to evaluate the performance of the models in both the training and validation subsets. 

The final models were also computed using stepwise, backward selection, multivariable logistic regression with a threshold of 0.10 for removal and a maximum of 20 iterations. However, unlike the cross-validation, all the available data (100 patients) were considered to train the model. All the variables that emerged as potential candidates (*p* < 0.1) in any univariate analysis of any iteration of the cross-validation were introduced as independent variables or predictors. The evaluation metrics of the final model were computed as the average of the evaluation metrics of the 5 models obtained during the cross-validation, for both the training and validation subsets.

Comparisons between variables were performed using IBM SPSS Statistics version 22 (IBM, New York, NY, USA). Regressions and cross-validation were performed using MATLAB (MathWorks, Natick, MA, USA). Statistical significance was set at *p* < 0.05 for all analyses.

## 3. Results

### 3.1. Participants

A total of 100 patients (29 women and 71 men,) met the inclusion criteria and were, consequently, included for analysis. Patients had a mean age and standard deviation of 37.7 ± 18.0 years. Fourteen patients from the total were less than 18 years old. Patients were admitted with UWS due to a traumatic (*n* = 40) or non-traumatic brain injury (*n* = 60) that occurred a mean of 132.8 ± 85.5 days prior to admission. At admission, 77 patients had a time since injury of less than 6 months, 40 of whom had a time since injury of less than 3 months.

Over the course of the analyzed period, 34% of the patients progressed to MCS, 12% of the patients emerged from MCS and 10% of the patients died (Figure 1). Changes of the neurobehavioral condition are described in detail below. 

The same group of participants was included for analysis in an accompanying paper about the functional independence of patients with UWS [53]. 

### 3.2. Transition to Minimally Conscious State

Thirty-four patients (9 women and 25 men) with a mean age and standard deviation of 33.6 ± 14.9 years progressed from UWS to MCS after a mean time since onset of 178.3 ± 107.3 days, and showed a mean score on the CRS-R of 9.8 ± 2.3 (Table 1). Transition to MCS occurred in the first 3 months post-injury in 6 patients, from 3 to 6 months post-injury in 19 patients, from 6 to 9 months in 5 patients, from 9 to 12 months in one patient, and after 12 months in 3 patients. 

Patients who transitioned from UWS to MCS were more likely to have suffered a traumatic injury (*Χ*^2^ (1, *N* = 100) = 10.168, *p* = 0.001), had shorter time since injury (*t*(98) = 2.421, *p* = 0.017), and had higher scores on the CRS-R at admission (*t*(98) = −3.536, *p* = 0.001). No differences in age, sex or education at admission were detected.

Transition to MCS was denoted by behavioral changes in a single domain of the CRS-R in 23 patients (67.6%), in two domains in 10 patients (29.4%), and in three domains in one patient (2.9%) (Figure 2 and Figure 3). Among all the patients who transitioned to MCS by showing signs in a single domain, 17 (73.9%) showed a visual sign, 5 (21.7%) showed a motor sign, one (4.3%) showed an auditory sign, and the remaining patient (4.3%) showed a communication sign. Visual signs, either fixation or visual pursuit, were the most common behavioral signs that denoted MCS and were exhibited by 28 of the 34 patients (82.3%). The most frequent combination of signs that denoted MCS involved visual and motor responses. 

The univariate logistic regressions of the cross-validation featured age as a potential predictor in two iterations, and etiology, time since injury and total score on the CRS-R in all five iterations. All these variables were consequently included in the multivariable logistic regression of the final model but only etiology, time since injury and total score on the CRS-R proved to be significant predictors. The *p*-value of the likelihood-ratio test of all the regression models was <0.001. Etiology was the most important variable contributing to predict the transition to MCS, followed by the score on the CRS-R (Table 2). Specifically, patients with traumatic injuries were almost five times more likely to transition to MCS than patients with non-traumatic injuries, and one-point increase in the CRS-S at admission doubled the likelihood of transition to MCS. Although the influence of time since injury on the final model was significant, the odds ratio of this factor showed that either it was not actually associated with transition to MCS or the association was weak.

As hypothesized, the performance of the model in the training subsets was better than in validation subsets, while differences were not significant (Table 2). The AUC values, greater than 0.8 in both the training and validation subsets supported the good performance of the final model. Regarding performance on new data, the model showed an accuracy to identify patients who did and did not transitioned to MCS of 75%. Transition to MCS occurred in 67% of the patients who were predicted to transition. From all the patients who actually transitioned to MCS, the model only predicted 56% of them correctly. In contrast, the model properly detected 85% of the patients who remained in UWS.

### 3.3. Emergence from Minimally Conscious State

Twelve patients with UWS at admission, 11 men and one woman, progressed to MCS and emerged from this state during the analyzed period (Table 3). At emergence, patients had a mean age of 24.2 ± 6.9 years, a mean time since injury of 251.1 ± 160.6 days, and a mean score on the CRS-R of 19.1 ± 2.9. Patients emerged from MCS after a mean time in MCS of 91.8 ± 60.6 days. Only two of them (16.7%) were in MCS for less than one month before emerging from this state. Five patients (41.7%) emerged from MCS in the first 6 months after the injury, 4 patients (33.3%) emerged from 6 to 9 months after the injury, one patient (8.3%) emerged from 9 to 12 months after the injury, and the remaining 2 patients (16.7%) emerged after one year from the injury (Figure 1). 

Patients who emerged from MCS were younger than those who did not (*t*(98) = 2.805, *p* = 0.006), were more likely to have suffered a traumatic injury (*Χ*^2^ (1, *N* = 100) = 10.669, *p* = 0.001), and had higher scores on the CRS-R at admission (*t*(98) = −2.504, *p* = 0.014) (Table 4). No differences in sex, education or time post-injury at admission were detected. 

Nine of the 12 patients who emerged from MCS (75%) showed one single behavioral sign, either functional communication or functional object use (Figure 4). The remaining three patients showed both signs.

The univariate logistic regressions of the cross-validation featured the score on the CRS-R at admission as a potential predictor in four iterations, and age and etiology in all five iterations. The three variables were consequently included in the multivariable logistic regression of the final model. Both etiology and score on the CRS-R emerged as significant predictors of emergence from MCS, while age showed a borderline significance (Table 5). The *p*-value of the likelihood-ratio test of all the regression models was <0.001. 

However, although the model had an AUC of 0.78 and an accuracy of 88% to predict emergence from MCS in new datasets, which would indicate a good performance of the model, these results were misleading. The precision and sensitivity of the model showed that only 10% of the patients who were predicted to emerge from MCS, actually did and, additionally, the model only detected 10% of the patients who did emerge. The model, conversely, detected 99% of the patients who did not emerge. These performance metrics evidenced that the model underrepresented the likelihood of emergence and turned out to predict that almost all patients would not emerge, which explains the high sensitivity and poor precision and sensitivity. Although the accuracy of the model was remarkable, as most patients did not emerge from MCS, and both the model and the identified predictors were found to be statistically significant, this model could not be accounted for.

### 3.4. Mortality

Ten patients died over the period analyzed, either from complicated infectious processes that led to septic conditions, as registered in eight cases (80%), or cardiorespiratory arrest, as registered in the remaining two cases (20%). No deaths were caused by withdrawal of nutritional or hydration support. Mean time from injury to death was 376.7 ± 291.5 days. Seven patients (70%) died during the first year after the injury and the remaining three patients (30%) died after this period. At the time of death, two patients (20%) had progressed to the MCS and had a mean score on the CRS-R of 12.5 ± 0.7. The remaining patients (80%) were in UWS and had a mean score on the CRS-R of 6.1 ± 1.4 at the time of death. 

Deceased patients were older than those who survived (*t*(98) = −3.846, *p* < 0.001)(Table 6). No other significant differences were found between these groups in either sex (*Χ*^2^(1, *N* = 100) = 0.653, *p* = 0.419), etiology (*Χ*^2^(6, *N* = 100) = 1.780, *p* = 0.939), time since injury at admission (*t*(98) = −0.112, *p* = 0.911), CRS-R at admission (*t*(98) = 1.764, *p* = 0.081), or education (*t*(98) = 98, *p* = 0.295).

The univariate logistic regressions of the cross-validation featured the score on the CRS-R at admission as a potential predictor in two iterations, and age in all five iterations. Both variables were consequently considered in the final model, which only identified age as a significant predictor for death (Table 7). The *p*-value of the likelihood-ratio test of all the regression models was <0.001.

Analogously to the prediction of emergence from MCS, although the resulting predictive model of death had an AUC of 0.78 and an accuracy of 90% in new data, the precision and sensitivity of the model evidenced that only 20% of the patients who were predicted to die actually did, and the model only predicted correctly 10% of the patients who died during the analyzed period. According to its specificity, the model underrepresented the likelihood of death and predicted that almost all the patients survived, which, unfortunately, was not the case. Again, although the model and the identified predictor were statistically significant and the accuracy of the model was remarkable, this model could not be accounted for.

## 4. Discussion

The present study describes the neurobehavioral course of a cohort of 100 patients with UWS consecutively admitted to a multidisciplinary rehabilitation unit and investigates predictors of transition to and emergence from MCS and mortality using a cross-validation methodology. According to our results, during the analyzed period, one tenth of the patients with UWS died, near one third of the patients were able to progress from UWS to MCS, and near one tenth of the total sample (near one third from those who progressed to MCS) were able to emerge from MCS. Transition to MCS was mostly denoted by visual signs, either fixation or visual pursuit, which commonly appeared alone or in combination with motor signs, and was predicted by etiology and the score on the CRS-R at admission with an accuracy of 75%. Emergence from MCS was denoted likewise by functional communication and object use. Predictive models of emergence from MCS and mortality were not valid and the identified predictors could not be accounted for. The functional independence of this cohort of patients is described in an accompanying paper [53].

Since the description of the diagnostic criteria of the MCS in 2002 enabled identifying this neurobehavioral condition, a series of studies have documented the neurobehavioral progress of patients with the current understanding of UWS. However, differences in the methodology and included samples among studies challenge comparison of their findings. The results of the latest studies with representative samples seem to evidence that around half of the patients who are admitted in a UWS, remain in this condition 6 months after the injury [23,34]. The higher proportion of patients in UWS at admission in our study by that time could be explained by the fact that patients were admitted to our facilities, and our study, with up to one year post-injury, while only patients with less than three months of evolution were included in the other studies [23,34]. However, transition to MCS at 6 months since injury was detected, as in our study, in almost one third of patients [23,34]. The higher likelihood of transition to MCS by the identification of one single sign detected in our study is supported by previous reports [42,45]. Furthermore, in accordance with our findings, visual signs, either visual pursuit and fixation, have been detected in the great majority of patients who transition to MCS, with a very similar proportion to our results, followed by motor responses [42]. In line with this, improvement in the CRS-R at 6 months post-injury has been reported to be mainly promoted by visual and motor responses [37]. The concurrent presence of visual and motor signs at transition to MCS has been also reported to be the most common combination of neurobehavioral signs [42]. The probability of emergence from MCS detected in our study is supported by previous results, which describe recovery of consciousness in a variable range of patients that range from almost 10% [34] to 18% [23]. Information about signs denoting emergence from consciousness is, however, scant and describes the appearance of functional object use in three patients [45] and the combination of both functional communication and object use in one patient [42]. Although our study provides the most comprehensive description of the neurobehavioral signs of emergence from MCS to date, the limited number of patients might restrict extrapolation of the results. The neurobehavioral improvement detected in our study is supported by a previous study, which reported an improvement in 38.9% of the patients who were admitted in UWS [29]. However, the mortality detected in our study, which is in accordance with previous studies [34,38], is dramatically lower. The higher incidence of death reported by this study could be partially explained by the lower time post-injury at admission and, to a greater extent, the high percentage of patients with vascular and anoxic etiology [29]. 

Etiology, in favor of patients with a traumatic brain injury, and the score on the CRS-R at admission, with higher scores being associated with better outcome, have been identified as significant factors in the neurobehavioral recovery. Since the work of the Multi Society Task Force [1], almost all existing studies have reported more favorable clinical course in patients with DOC of traumatic origin compared to other etiologies [8,27,28,30,35,47]. The recommendations of the American and European Academy of Neurology, highlighting the sensitivity of the CRS-R to detect signs of consciousness have contributed to its widespread use [3,44]. A remarkable number of studies have reported its usefulness to predict an improvement of the neurobehavioral condition [29,33,34,41], including an emergence from the MCS [35,38,47]. Our results supported the predictive value of both etiology and baseline score on the CRS-R to identify those patients with UWS with higher change to transition to MCS. However, performance of the predictive model was moderate. Contrary to previous studies, which evaluate the performance of the predictive models with the same data that was used to create the model [23,29,33,35,37,38,39,46,47,50], we used a cross-validation methodology to have a realistic estimation of the performance of the model on new data; it is, to determine the performance of the model on new patients admitted to our facilities. As hypothesized, our models had worse performance when new data were considered, which supports the possibility that results of other studies might be somewhat optimistic. However, and more importantly, studies rarely provide a reasonable choice of metrics that allow interpretation of the performance of the model [23,29,33,34,35,38,41,47]. Rather, most studies provide accuracy as the most relevant, if not the only, performance metric. While this measure can be self-explanatory, relying on a single measure like accuracy can be misleading. The limited patient records with the current definition of UWS, and systematic assessments with standardized instruments, as the CRS-R, lead to highly unbalanced data. The performance of classification methods on these data should rely on several measures, such as precision, sensitivity and specificity, in addition to accuracy [55]. Particularly in our study, although the predictive models of emergence from MCS and mortality were nearly 90% accurate and the model and the identified predictors were statistically significant, the precision, sensitivity and specificity of the models evidenced that these models were not correct. Specifically, the models were underrepresenting the likelihood of emergence from MCS and death; it is, they were predicting that almost no patient would emerge or die, which turned out to be nearly 90% accurate, as only around 10% of patients actually emerged from MCS or died, but useless in terms of prediction. Increasing the data through multicenter studies would not presumably solve the data imbalance, but would certainly enable addressing imbalance problems with increased robustness, improve the validity of the results and allow for investigating the classification problem with other techniques and classifiers. The joint effort of Estraneo and colleagues from the International Brain Injury Association’s DOC Special Interest Group should be highlighted and could be the way forward [34]. 

All our findings should be interpreted taking into account the characteristics of our sample. First, all patients included in our study were provided multidisciplinary rehabilitation in a specialized neurorehabilitation center, which could restrict the generalization of our results to other populations without access to rehabilitation resources. Second, it should be considered that time since injury to admission was variable and was higher than three months in half of our patients. Although these values of time since injury may be representative of patients with UWS at admission to neurorehabilitation centers, it should be taken into account that time post-injury may influence both the percentage of patients who emerge from MCS or die [3,29,44,56]. Third, the assessment of the neurobehavioral condition was exclusively done with the CRS-R, which might have limited sensitivity to detect the ability of patients with impaired motor output to show some degree of consciousness [57]. Although the addition of neurophysiological or neuroimaging examinations in the assessment protocol could have improved the detection of command following [58,59], the restricted access to instrumented tests in comparison to bedside instruments, could have limited the extrapolation of the findings. It is important to highlight that the CRS-R is considered the reference standard (in the absence of a gold standard) of the clinical bedside evaluation for signs of consciousness [3,44]. Further, in line with these recommendations, we considered that the identification of visual fixation in patients with UWS denoted transition to MCS. However, the reliability of visual fixation to reflect consciousness and higher order cortical brain function is controversial [60]. Fourth, the neuropathological characteristics of the brain injuries, such as the localization and side of the lesion, were not considered in the models, as different etiologies with different neuroanatomical bases were included. This information could be especially relevant and worthy to be explored when investigating the neurobehavioral progress of a specific etiology of brain injury. Finally, we assume linear relationships between the potential predictors and the neurobehavioral changes, which might not be the case. Investigating the neurobehavioral progress of patients with UWS with other machine learning techniques could yield different results. However, the systematic weekly assessments of the neurobehavioral condition conducted in our study using the CRS-R and the validation and reasonable interpretation of the predictive models support the reliability of the clinical course detected during the analyzed period and the validity of the identified predictors. 

## 5. Conclusions

Systematic assessment of the clinical course of 100 patients with UWS and a time since injury of 132.8 ± 85.5 days using standardized and specific measures showed that near one third of the patients admitted with UWS were able to progress to MCS, near one tenth of them were able to emerge from MCS, and one tenth of the total sample died. Transition to MCS was mostly denoted by visual signs, either fixation or visual pursuit, which appeared alone or in combination with motor signs, and was predicted by etiology and the score on the CRS-R at admission with an accuracy of 75%. Emergence from MCS was denoted by functional communication and object use in the same proportion. Predictive models of emergence from MCS and mortality were not valid and the identified predictors could not be accounted for.

## Figures and Tables

**Figure 1 brainsci-11-00126-f001:**
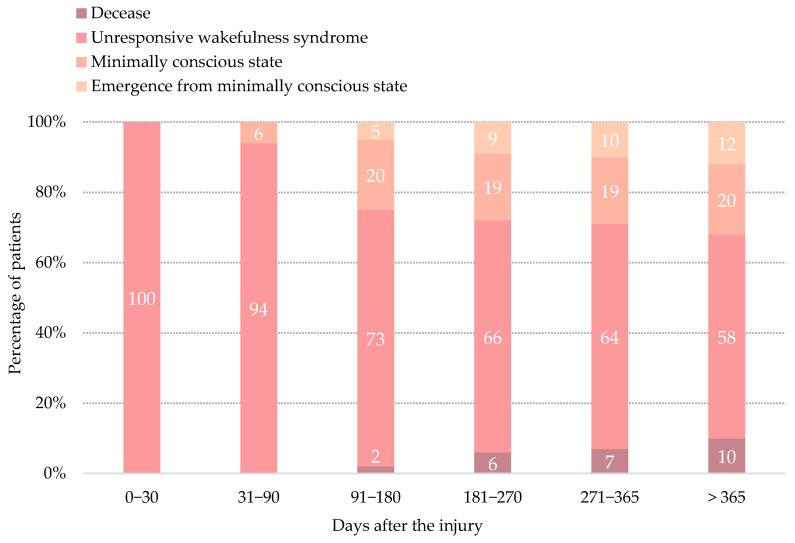
Clinical and neurobehavioral progress of the included cohort.

**Figure 2 brainsci-11-00126-f002:**
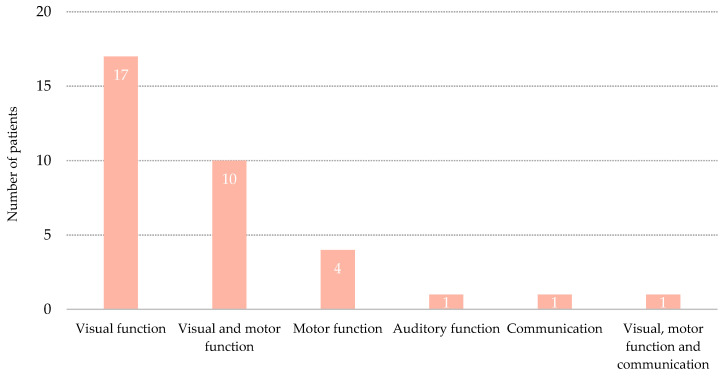
Distribution of patients per domains of the Coma Recovery Scale that denoted Minimally Conscious State.

**Figure 3 brainsci-11-00126-f003:**
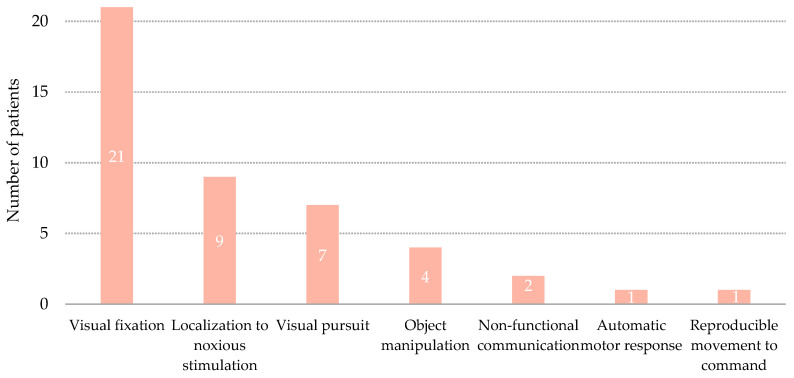
Distribution of patients per items of the Coma Recovery Scale that denoted Minimally Conscious State.

**Figure 4 brainsci-11-00126-f004:**
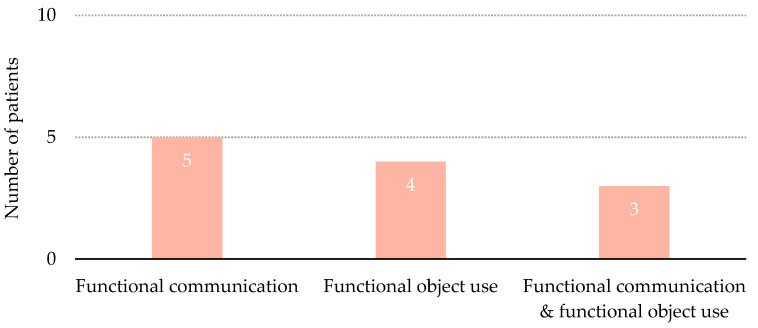
Distribution of patients per items of the Coma Recovery Scale that denoted emergence from Minimally Conscious State.

**Table 1 brainsci-11-00126-t001:** Baseline demographic and clinical characteristics of the patients who did and did not transition to Minimally Conscious State.

	Patients Who Transitioned to MCS (*n* = 34)	Patients Who Remained in UWS (*n* = 66)	Significance
Age (years)	33.6 ± 15.0	39.8 ± 19.2	NS (*p* = 0.100)
Sex (*n*, %) Women Men	9 (26.5%) 25 (73.5%)	20 (30.3%) 46 (69.7%)	NS (*p* = 0.689)
Education (years)	10.9 ± 4.4	10.8 ± 5.4	NS (*p* = 0.908)
Etiology (*n*, %) Traumatic Non-traumatic	21 (61.8%) 13 (38.2%)	19 (28.8%) 47 (71.2%)	*p* = 0.001
Time since injury (days)	104.6 ± 64.0	147.3 ± 91.9	*p* = 0.017
Coma Recovery Scale-Revised	6.3 ± 1.4	5.4 ± 1.2	*p* = 0.001

Age, time since injury and score on the Coma Recovery Scale refer to values at admission. Age, education, time since injury and score on the Coma Recovery Scale-revised are expressed in terms of mean ± standard deviation. NS: not significant.

**Table 2 brainsci-11-00126-t002:** Parameters and validation metrics of the predictive model of transition to Minimally Conscious State.

Multivariable Logistic Regression Model				Cross-Validation Evaluation Metrics		
	B (SE)	OR (95% CI)	*p*		Training	Validation
				Accuracy (95% CI)	0.79 (0.77–0.80)	0.75 (0.67–0.83)
Constant	−4.27 (1.30)	0.01 (0.00–0.18)	0.001	Precision (95% CI)	0.73 (0.70–0.76)	0.67 (0.54–0.79)
Etiology ^1^	1.57 (0.51)	4.78 (1.77–12.90)	0.002	Sensitivity (95% CI)	0.60 (0.53–0.68)	0.56 (0.36–0.76)
Time since injury	−0.01 (0.01)	0.99 (0.98–1.00)	0.007	Specificity (95% CI)	0.88 (0.86–0.91)	0.85 (0.76–0.94)
Coma Recovery Scale-Revised	0.70 (0.22)	2.01 (1.32–3.08)	0.001	AUC (95% CI)	0.82 (0.80–0.84)	0.80 (0.76–0.84)

^1^ Traumatic *vs* non-traumatic. AUC: Area under de ROC curve. B: beta. SE: standard error. OR: odds ratio. CI: confidence interval.

**Table 3 brainsci-11-00126-t003:** Individual characteristics and neurobehavioral progress until emergence from the minimally conscious state.

Patients	Characteristics		Admission			Progression from UWS to MCS					Emergence from MCS				
	Sex	Etiology	Age	Time Since Injury	CRS-R	Age	Time Since Injury	Time Since Admission	CRS-R	Signs	Age	Time Since Injury	Time Since Admission	CRS-R	Signs
Patient 1	M	Traumatic	25	86	5	25	110	24	6	V	26	271	185	20	C
Patient 2	M	Anoxia	14	38	7	14	76	38	10	V	14	104	66	22	C
Patient 3	M	Traumatic	26	38	6	26	76	38	18	M	26	132	94	23	C/FU
Patient 4	M	Traumatic	19	240	3	20	486	246	14	V/M	20	668	428	20	C
Patient 5	W	Traumatic	31	146	7	31	178	32	8	V	32	346	200	19	C/FU
Patient 6	M	Traumatic	31	62	8	31	89	27	10	M	31	172	110	18	C/FU
Patient 7	M	Traumatic	25	213	8	25	250	37	10	V/M	25	401	188	16	C
Patient 8	M	Traumatic	33	45	6	33	91	46	11	V/M	33	201	156	19	FU
Patient 9	M	Fat embolism	19	45	6	19	86	41	10	C	19	99	54	12	C
Patient 10	M	Traumatic	21	117	8	21	181	64	9	V	21	243	126	21	FU
Patient 11	M	Traumatic	13	126	8	13	158	32	12	V	13	222	96	21	FU
Patient 12	M	Traumatic	31	101	7	31	130	29	9	V	31	154	53	18	FU

Age is expressed in years. Time since injury and time since admission are expressed in days. M: man. W: woman. V: visual. C: communication. M: motor. FU: functional use.

**Table 4 brainsci-11-00126-t004:** Baseline demographic and clinical characteristics of the patients who did and did not emerge from Minimally Conscious State.

	Patients Who Emerged from MCS (*n* = 12)	Patients Who Did not Emerge from MCS (*n* = 88)	Significance
Age (years)	24.5 ± 6.8	39.5 ± 18.4	*p* = 0.006
Sex (*n*, %) Women Men	1 (8.3%) 11 (91.7%)	28 (31.8%) 60 (68.2%)	NS (*p* = 0.093)
Education (years)	10.2 ± 2.6	10.9 ± 5.3	NS (*p* = 0.690)
Etiology (*n*, %) Traumatic Non-traumatic	10 (83.3%) 2 (16.7%)	30 (34.1%) 58 (65.9%)	*p* = 0.001
Time since injury (days)	105.2 ± 68.7	136.6 ± 87.2	NS (*p* = 0.236)
Coma Recovery Scale-Revised	6.6 ± 1.5	5.6 ± 1.3	*p* = 0.014

Age, time since injury and score on the Coma Recovery Scale refer to values at admission. Age, education, time since injury and score on the Coma Recovery Scale-revised are expressed in terms of mean ± standard deviation. NS: not significant.

**Table 5 brainsci-11-00126-t005:** Parameters and validation metrics of the predictive model of emergence from Minimally Conscious State.

Multivariable Logistic Regression Model				Cross-Validation Evaluation Metrics		
	B (SE)	OR (95% CI)	*p*		Training	Validation
				Accuracy (95% CI)	0.89 (0.87–0.90)	0.88 (0.86–0.90)
Constant	−4.76 (2.10)	0.01 (0.00–0.53)	0.024	Precision (95% CI)	0.50 (0.06–0.94)	0.10 (0.00 *−0.30)
Age	−0.07 (0.04)	0.93 (0.87–1.00)	0.050	Sensitivity (95% CI)	0.08 (0.00–0.16)	0.10 (0.00 *–0.30)
Etiology ^1^	2.04 (0.86)	7.68 (1.42–41.52)	0.018	Specificity (95% CI)	1.00 (0.99–1.00)	0.99 (0.97–1.00 *)
Coma Recovery Scale-Revised	0.58 (0.29)	1.79 (1.02–3.14)	0.041	AUC (95% CI)	0.86 (0.84–0.88)	0.78 (0.68–0.89)

^1^ Traumatic vs non-traumatic. AUC: Area under de ROC curve. B: beta. SE: standard error. OR: odds ratio. CI: confidence interval. * Corrected probability to be between (0–1) values.

**Table 6 brainsci-11-00126-t006:** Baseline demographic and clinical characteristics of the patients who did and did not emerge from Minimally Conscious State.

	Patients Who Died (*n* = 10)	Patients Who Survive (*n* = 90)	Significance
Age (years)	57.2 ± 15.6	35.5 ± 17.0	*p* < 0.001
Sex (*n*, %) Women Men	4 (40.0%) 6 (60.0%)	25 (27.8%) 65 (72.2%)	NS (*p* = 0.419)
Education (years)	12.4 ± 5.9	10.6 ± 5.0	NS (*p* = 0.295)
Etiology (*n*, %) Traumatic Non-traumatic	4 (40.0%) 6 (60.0%)	36 (40.0%) 54 (60.0%)	NS (*p* = 1.000)
Time since injury (years)	135.7 ± 110.1	132.5 ± 83.1	NS (*p* = 0.911)
Coma Recovery Scale-Revised	5.8 ± 1.3	5.0 ± 1.3	NS (*p* = 0.081)

Age, time since injury and score on the Coma Recovery Scale refer to values at admission. Age, education, time since injury and score on the Coma Recovery Scale-revised are expressed in terms of mean ± standard deviation. NS: not significant.

**Table 7 brainsci-11-00126-t007:** Parameters and validation metrics of the predictive model of death.

Multivariable Logistic Regression Model				Cross-Validation Evaluation Metrics		
	B (SE)	OR (95% CI)	*p*		Training	Validation
				Accuracy (95% CI)	0.91 (0.90–0.91)	0.90 (0.87–0.93)
				Precision (95% CI)	0.60 (0.12–1.00 *)	0.20 (0.00 *–0.59)
				Sensitivity (95% CI)	0.08 (0.02–0.14)	0.10 (0.00 *–0.30)
Constant	−5.78 (1.37)	0.00 (0.00–0.05)	<0.001	Specificity (95% CI)	1.00 (1.00–1.00)	0.99 (0.97–1.00 *)
Age	0.08 (0.02)	1.08 (1.03–1.10)	0.002	AUC (95% CI)	0.83 (0.79–0.87)	0.78 (0.61–0.94)

AUC: Area under de ROC curve. B: beta. SE: standard error. OR: odds ratio. CI: confidence interval. * Corrected probability to be between (0–1) values.

## Data Availability

The data presented in this study are available on request from the corresponding author. The data are not publicly available due to privacy and ethical restrictions.

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
