# Peer review of "When, How, and to What Extent Are Individuals with Unresponsive Wakefulness Syndrome Able to Progress? Neurobehavioral Progress"

_brainsci, 2021, doi:10.3390/brainsci11010126_

Round 1

Reviewer 1 Report

Overall assessment:

This is an interesting article worthy of publication with some revisions. The strengths of this work include the sample size (relatively large for this domain) and the frequency of behavioural evaluations (more assessments over a longer period of time than most other work in this domain).

Major comment:

A consideration that is missing in the determination of UWS vs MCS is attributing the diagnosis of MCS using visual fixation. Bruno and colleagues (1) provide evidence that sustained visual fixation does not necessarily reflect consciousness and higher order cortical brain function. This corresponds with debate in the older clinical literature; briefly, the Multi Society Task Force on PVS and UK guidelines (2, 3) support that visual fixation can be demonstrated in the UWS, while the Aspen Neurobehavioral Conference uses fixation as a clinical criterion defining MCS (4). Where do the authors stand on this issue? This is highly relevant to the results, given that visual fixation was the most commonly observed behaviour in their MCS cohort, and this consideration deserves attention in the manuscript as it would very likely substantially change their results.

On a related note, Candelieri and colleagues (5) found that the detection of visual pursuit was variable depending on the time of day of testing and much less frequently detected in post-prandial assessments. At what time of day were the patients assessed, and to what extent do the authors suspect their results are influenced by the patients' circadian variability?

Minor comment:

Line 406 - "Thirds" is a typographical error; it should be "Third".

References

  1. Bruno M-A, Vanhaudenhuyse A, Schnakers C, Boly M, Gosseries O, Demertzi A, et al. Visual fixation in the vegetative state: An observational case series PET study. BMC Neurol. 2010;10:35.
  2. Multi-Society Task Force on PVS. Medical aspects of the persistent vegetative state (first part). N Engl J Med. 1994;330:1499–508.
  3. Working Party of the Royal College of Physicians. The vegetative state: Guidance on diagnosis and management. Clin Med. 2003;3:249–54.
  4. Giacino JT, Ashwal S, Childs NL, Cranford R, Jennett B, Katz DI, et al. The minimally conscious state: Definition and diagnostic criteria. Neurology. 2002;58:349–53.
  5. Candelieri A, Cortese MD, Dolce G, Riganello F, Sannita WG. Visual pursuit: Within-day variability in the severe disorder of consciousness. J Neurotrauma. 2011;28:2013–2017.

Reviewer 2 Report

The authors report about a cohort of 100 patients with post severe BI UWS to describe the predictors of evolution towards MCS or E-MCS.

Near one third of the patients were able to progress to MCS, near one tenth of them were able to emerge from MCS, and one tenth died.

Transition to MCS was mostly denoted by visual signs (assessedby CRS-R) and was predicted by etiology and the score in the CRS-R at admission.

The article is interesting and touches a relevant subject which have implications in the clinical practice (allocating resources, withdrawal decisions, etc.). However I see a major limitation that should be addressed and extensively discussed. All the observations are based on the assumption that CRS-R represents the most valid and sensitive scale to diagnose UWS.

Roughly 15-20% of patients with a clinical diagnosis of UWS show some sort of commands following on paraclinical testing (either f-MRI or EEG) indeed the diagnose of UWS based on clinical evaluation presents an intrinsic bias being potentially confused with the cognitive motor dissociation condition (which has a completely different clinical evolution over time).

The CRS-R, which is mainly based on residual motor output, in case of impaired motor efference/output, in the case of lesions affecting strategic functional areas of the central nervous system, or processes altering the functionality of the peripheral nervous system is probably unfit to unmask patient’s cognitive abilities to interact.

This is a crucial point, therefore more data should be presented to differentiate patients evolving to MCS (localization of lesions, side, comorbidities, etc.) and these data should be included in the models. 

The article is hard to read because it’s too long; a more concise introduction and presentation of results is recommended.

Reviewer 3 Report

Thank you for the opportunity to review the manuscript titled “When, how and to what extent are individuals with unresponsive wakefulness syndrome able to progress? Neurobehavioral progress”. This prospective cohort study describes the neurobehavioural course of 100 patients with unresponsive wakefulness syndrome who were consecutively admitted to a multidisciplinary rehabilitation unit. It examines predictors of transition to and emergence from minimally conscious states and mortality during 12 months post-injury using cross-validation methodology. The study is reasonably well conducted, especially considering the challenges with investigations of this patient cohort. I offer a few comments for the authors consideration.

  1. L13-27: Please define all abbreviations on first use in the abstract. The reader should not have to refer to the main body to fully understand the abstract.
  2. L20-22: Report exact frequencies and proportions of patients rather than ’one-tenth’.
  3. L141-142: Please replace ‘quantitative’ and ‘qualitative’ with the more customary ‘continuous’ and ‘categorical’ (or ‘nominal’) labels, respectively.
  4. L143-182: I believe the authors actually mean to say ‘multivariable’, not ‘multivariate’. Please see https://www.ncbi.nlm.nih.gov/pmc/articles/PMC3518362/ and confirm.
  5. L193-194: Report exact frequencies and proportions of patients rather than ‘near one third’ and ’near one tenth’.
  6. L207-210: I suggest removing ‘NS’ and ‘p=’ in the last column and simply reporting the p value. The column heading can be changed to ‘P value’.
  7. L211-224: It is customary to report both frequencies and proportions. Normally, I would insist on it, but understand this may feel somewhat redundant when the total sample size is 100. At least report both frequencies and proportions when reporting on patients from subgroups (i.e. patients who transitioned to MCS). This comment applies to relevant parts of the rest of the result section also.
  8. L244-246: Table 2 is really two separate tables joined together. The ‘first’ table (i.e. the regression model output) includes unnecessary information. Although I appreciate the completeness of the reporting, the only useful information is the odds ratios with 95% confidence intervals for the three variables. These could be reported in the manuscript text only.
  9. L260-263: I suggest removing ‘NS’ and ‘p=’ in the last column and simply reporting the p value. The column heading can be changed to ‘P value’.
  10. L286-289: Table 5 is really two separate tables joined together. The ‘first’ table (i.e. the regression model output) includes unnecessary information. Although I appreciate the completeness of the reporting, the only useful information is the odds ratios with 95% confidence intervals for the three variables. These could be reported in the manuscript text only.
  11. L302-305: I suggest removing ‘NS’ and ‘p=’ in the last column and simply reporting the p value. The column heading can be changed to ‘P value’.
  12. L318-319: Table 2 is really two separate tables joined together. The ‘first’ table (i.e. the regression model output) includes unnecessary information. Although I appreciate the completeness of the reporting, the only useful information is the odds ratios with 95% confidence intervals for the three variables. These could be reported in the manuscript text only.

Round 2

Reviewer 2 Report

-

Author Response

We thank the Reviewer 2 for the time spent in our manuscript